# Modulation of Paracellular-like Drug Transport across an Artificial Biomimetic Barrier by Osmotic Stress-Induced Liposome Shrinking

**DOI:** 10.3390/pharmaceutics14040721

**Published:** 2022-03-28

**Authors:** Jonas Borregaard Eriksen, Hesham Barakat, Barbara Luppi, Martin Brandl, Annette Bauer-Brandl

**Affiliations:** 1Department of Physics Chemistry and Pharmacy, University of Southern Denmark, Campusvej 55, DK-5230 Odense, Denmark; borregaard@sdu.dk (J.B.E.); hesham.barakat@studio.unibo.it (H.B.); annette.bauer@sdu.dk (A.B.-B.); 2Department of Pharmacy and Biotechnology, Alma Mater Studiorum—University of Bologna, Via San Donato 19/2, 40127 Bologna, Italy; barbara.luppi@unibo.it

**Keywords:** biophysical models, biomimetics, intestinal absorption, in vitro models, liposomes, osmotic pressure, paracellular transport, passive diffusion, permeability, permeability coefficients

## Abstract

Various types of artificial biomimetic barriers are widely utilized as in vitro tools to predict the passive “transcellular” transport of drug compounds. The current study investigated if the sandwich barrier PermeaPad^®^, which is composed of tightly packed phospholipid vesicles enclosed between two support sheets, contributes to a transport mechanism that is paracellular-like, representing one of the alternative pathways of passive transport in vivo, primarily of relevance for hydrophilic drug compounds. To this end, we pretreated the commercial PermeaPad^®^ barrier with NaCl solutions of either high or low osmolality prior to permeation experiments on reversed Franz cell setups with hydrophilic model compounds calcein and acyclovir and hydrophobic model compounds hydrocortisone and celecoxib. Our starting hypothesis was that the liposomes formed in the barrier during the hydration step should either shrink or swell upon contact with test media (drug solutions) due to osmotic pressure difference as compared to the pretreatment solutions. Apparent permeabilities for calcein and acyclovir across the PermeaPad^®^ barrier were found to increase approximately 2.0 and 1.7 fold, respectively, upon hypo-osmotic pretreatment (soaking in hypotonic medium, while the permeation of hydrocortisone and celecoxib remained unchanged. A control experiment with lipid-free barriers (support sheets) indicated that the permeation of all the compounds was virtually unchanged upon hypo-osmotic pretreatment. In conclusion, soaking PermeaPad^®^ in a medium of lower osmotic pressure than that used during the permeation study appears to induce the osmotic shrinking of the lipid vesicles in the barrier, leaving wider water channels between the vesicles and, thus, allowing hydrophilic compounds to pass the barrier in a paracellular-like manner.

## 1. Introduction

To achieve therapeutic blood levels with orally administered drug compounds, they need to be absorbed from the gastrointestinal tract into systemic circulation. The main pathways for drug compounds passing across the intestinal barrier include active transport under energy consumption, endocytosis/transcytosis, and, most importantly, passive transport with a high transport capacity due to diffusion, along with concentration or chemical potential gradients. Thus, passive absorption is the rate-limiting route of absorption for the vast majority of drug compounds. Depending on their physicochemical characteristics, during passive transport, drugs may cross the endothelial cell itself (transcellular diffusion) or diffuse along the water channels between cells (paracellular diffusion). Lipophilic solutes tend to be more likely to cross biological barriers via the transcellular pathway, while hydrophilic molecules are said to pass biological barriers via the paracellular route [1].

In vitro drug transport studies aiming to predict intestinal absorption may utilize tissue-based, cell-based, or artificial barrier models. Depending on the type of cells and tissues used, the models may facilitate the study of active, passive paracellular, and passive transcellular transport, while artificial barriers are restricted to passive transport only. The gold standard for predicting paracellular transport in in vitro experiments is using human intestinal tissue, but it is not easy to acquire. Animal models are more readily accessible, but in many cases, their paracellular transport capacity is different from humans [2].

Cell-based models represent a more readily accessible in vitro approach. They consist of a monolayer of cells with tight junctions between cells similar to the cell layers in the intestine [3,4]. The Caco-2, MDCK II, and 2/4/A1 cell lines are three that are commonly used cell lines for such studies. Caco-2 and MDCKII cell models tend to underpredict the paracellular transport due to a lower porosity. At the same time, 2/4/A1 often obtains a similar paracellular permeation to the human intestine, although it has an entirely different pore size distribution [5]. Adson and coworkers proposed a mathematical model to correct the permeation values to predict the paracellular transport of molecules from the Renkin function [6]. The group of Surgano later successfully applied mathematical modeling in combination with artificial barrier studies [7].

The gold standard among non-cellular artificial barrier models, the PAMPA model, is commonly acknowledged to perform well for passive transcellular permeation studies due to its coherent layer of lipophilic liquid with dissolved/dispersed lipids. In contrast, the phospholipid vesicle-based permeation model (PVPA) and PermeaPad^®^ both represent artificial biomimetic barriers composed of phospholipid vesicles. In contrast to the PVPA model, where vesicles are preformed and deposited into filters, for the PermeaPad^®^ barrier, the dry lipids representing the middle layer of the sandwich (support–lipid–support) spontaneously form a tightly packed vesicular structure upon contact with aqueous solutions [8,9]. The morphology of the lipid layer of the PermeaPad^®^ barrier resembles semisolid vesicular phospholipid dispersions, also called vesicular phospholipid gels [10,11], which have earlier been described to form vesicular or multilamellar structures depending on phospholipid content and mechanical stress exerted during swelling. They were demonstrated to retain and slowly release drug molecules depending on their phospholipid content, i.e., how tightly the vesicles are packed [12,13]. Obviously, hydrophilic compounds may permeate via the gaps between the vesicles to cross the barrier’s lipid layer, similar to hydrophilic compounds permeating via the paracellular pathway of cellular barriers. To the best of our knowledge, there is only one study specifically investigating the paracellular-like transport across non-cellular (in comparison to cellular) barriers [3].

The hypothesis of the current study was connected to the assumption that the size of phospholipid vesicles will increase when exposed to an environment with a lower osmolality (a hypo-osmotic environment) than the environment in which the lipid vesicles were formed, and also the opposite, whereby the size will decrease in a medium with higher osmolality (hyper-osmotic environment) [14,15,16]. Due to the geometric constraints of the lipid layer in between the two support membranes within the PermeaPad^®^ barrier, the soaking of the barrier with hypo-/hyper-osmotic solutions (as compared to the tonicity of the drug solution) should influence their sizes and, in consequence, how tightly they are packed. In turn, an effect on the permeation of hydrophilic solutes should be observable.

In the present study, we tried to elucidate whether the transport of drug molecules through the gaps between lipid vesicles in the PermeaPad^®^ barrier plays a significant role. To this end, we investigated the transport of four model compounds of different lipophilicities (Table 1) using a reversed Franz cell setup across PermeaPad^®^, which had been pretreated under hypo-, iso-, and hyper-osmotic conditions.

## 2. Materials and Methods

### 2.1. Chemicals

Sodium phosphate dibasic dihydrate, sodium phosphate monobasic monohydrate, sodium hydroxide, calcein, and hydrocortisone (HPLC grade) were purchased from Sigma Aldrich^®^ Denmark ApS (Brøndby, Denmark). Acetonitrile (HPLC grade), trifluoroacetic acid, hydrochloric acid, and sodium chloride were purchased from VWR™ International A/S (Søborg, Denmark). Acyclovir (98%) and celecoxib were purchased from ABCR GmbH Germany (Karlsruhe, Germany). All water used for experiments and analytical purposes was analytical-grade, highly purified water prepared with a Milli-Q^®^ Reference A+ Water Purification System Merck KGaA (Darmstadt, Germany). All chemicals were of analytical grade unless stated otherwise.

### 2.2. Media Preparation

During these studies, phosphate-buffered saline (PBS) was used as the acceptor medium and for the preparation of donor media for permeation experiments. The liposomes were allowed to form within the barrier by soaking the dry sandwich barrier (PermeaPad) in hypo-, iso-, or hyper-osmotic media and compared to the PBS by using NaCl solutions of different osmolalities. PBS was prepared by dissolving 35 mM sodium phosphate dibasic dihydrate and 75 mM sodium phosphate monobasic monohydrate in water. Then, the pH was adjusted to 6.5 by adding 0.1 M HCl or 0.1 M NaOH. Finally, the osmolality was measured on a semi-micro osmometer K-7400 from KNAUER Wissenschaftliche Geräte GmbH (Berlin, Germany), and NaCl was added to obtain an osmolality of 300 mOsm. Solutions of NaCl were prepared by dissolving NaCl in purified water to obtain osmolalities of 50, 150, 250, 300, 350, 450 and 900 mOsm.

Solutions of acyclovir, calcein, and hydrocortisone and suspensions of celecoxib were the donor media in the permeation experiments. The acyclovir solution was prepared by dissolving 1 mg/mL acyclovir in PBS. The calcein solution was prepared by diluting a 58 mg/mL calcein stock solution at 1:20 in PBS to obtain a donor concentration of 2.9 mg/mL. The stock of calcein was prepared by dissolving 2.8 g calcein in 30 mL water before adjusting the pH to 6.5 with 4 M NaOH (~20 mL), and the osmolality of the stock was adjusted to 300 mOsm with NaCl. For hydrocortisone, a saturated suspension of 500 µg/mL hydrocortisone in PBS was filtrated through a hydrophilic 0.45 µm filter and then diluted in PBS to form a 250 µg/mL solution. The saturated celecoxib suspensions were prepared by suspending 1 mg/mL of celecoxib in PBS. The suspension was then sonicated for 10 min and left in the shaking water bath at 50 rpm and 37 °C for 40 h before starting the permeation experiments.

### 2.3. Permeation Experiments

For the permeation experiments, we used the reversed Franz cell setup from Permegear (Hellertown, PA, USA) with the donor in the bottom chamber (8 mL) and the acceptor in the top chamber (2 mL). The two compartments were either separated by PermeaPad^®^ or a sandwich consisting of two pieces of the support sheet representing the support layer of PermeaPad^®^ (with no lipids). The barriers were kindly donated by InnoMe GmbH (Espelkamp, Germany). A cylindrical stir bar (2 mm diameter × 8 mm length) was spinning at 500 rpm at the bottom of the donor compartment. The temperature of the Franz cells was kept constant at 37 °C by a circulating water system.

Prior to the permeation experiments, the barrier was exposed to NaCl solutions ranging from 50 to 900 mOsm for 30 min by simply adding 2 mL of the NaCl solution to the top compartment. The NaCl solutions were removed from the top compartment right before initiating the permeation experiments.

For the permeation experiments, the system was kept assembled after the NaCl pretreatment, and the bottom compartment was filled with 8 mL of donor solution/suspension, followed by the addition of 2 mL PBS to the top compartment. Then, 200 µL samples were withdrawn from the top compartment after 30, 45, 60, 75, and 90 min for hydrocortisone, acyclovir, and calcein, while for celecoxib, 600 µL samples were withdrawn after 120, 240, and 360 min. All samples were replaced by the same volume of PBS to keep the acceptor volume constant and to maintain sink conditions. Before and after experiments, donor samples were withdrawn to keep track of the mass balance (see the Appendix A).

### 2.4. Quantification

To quantify the amount of drug permeated in the permeation experiments, ultraviolet–visible spectroscopy (UV/VIS) was used for hydrocortisone and acyclovir, fluorescence spectroscopy was used for calcein, and high-performance liquid chromatography with ultraviolet detection (HPLC-UV) was used for celecoxib. Standard curves with a quantification limit of a lower concentration than the lowest concentrated samples were made for each measurement.

A BMG FLUOstar^®^ Omega microplate reader (BMG LABTECH, Ortenberg, Germany) was utilized to quantify hydrocortisone, acyclovir, and calcein. Hydrocortisone and acyclovir were detected with UV/VIS at 254 nm and 253 nm, respectively. Calcein was detected with fluorescence spectroscopy with the excitation and emission wavelengths at 485-12 and 520 nm, respectively.

Celecoxib was quantified with a 2695D HPLC apparatus from Waters Corporation (Milford, MA, USA) with a Dionex™ reversed-phase C18 LC-column from Thermo Fischer Scientific Inc. (Roskilde, Denmark) (product number: 059133). The column temperature was 40 °C during measurement. The mobile phase consisting of 65% acetonitrile and 35% of 0.1% trifluoroacetic acid in purified water was flowing through the system at a 1 mL/min rate. UV detection was at 254 nm with a 2487 Dual λ Absorbance detector from Waters Corporation coupled to the system. All quantification was with a freshly prepared calibration curve in the relevant range.

### 2.5. Data Analysis

The flux was determined by plotting the cumulative amount of a drug compound permeation (*dQ*) normalized to the permeation area (*A*) (1 cm^2^) against time (*dt*), with the flux (*J*) equal to the slope.
(1)$$J=dQ/\left(A\cdot dt\right)$$

We assumed that the change in the donor concentration over the experiments was negligible. The apparent permeability (*P_app_*) was equal to the flux normalized to the starting donor concentration (*C*_0_).
(2)$$P_{app}=\frac{J}{C_{0}}$$

The permeation across the lipid layer (*P_lip_*) has been calculated using Equation (3) (Di Cagno et al., 2015). The inverse of each permeability constant represents the resistance. The sum of the resistance of the lipid layer and the support layer (*P_sup_*) composing the PermeaPad^®^ barrier gives the resistance of the whole barrier.
(3)$$\frac{1}{P_{app}}=\frac{1}{P_{lip}}+\frac{1}{P_{sup}}$$

### 2.6. Statistics

Unpaired parametric t-tests with equal variances were utilized to determine significant differences between two means. For the data in Figure 1, one-factor ANOVA was used with Fischer’s LSD test for multiple comparisons of every mean to the mean of 300 mOsm as a control column. Statistical analysis was carried out in the GraphPad Prism 8.4.2 software. Differences were deemed significant for *p* < 0.05.

## 3. Results and Discussion

### 3.1. Permeation after Pretreatment with NaCl Solutions

Figure 1 shows the permeation of the four model compounds calcein, acyclovir, hydrocortisone, and celecoxib in PBS after the barrier had been pretreated with NaCl solutions of different osmolalities from 50 to 900 mOsm/kg. When looking at the hydrophilic compounds calcein and acyclovir, we observed that the permeabilities of these compounds in PBS increased by 1.5–2 fold when the barrier had been pretreated with NaCl solutions of 50, 150, or 900 mOsm/kg compared to when the barrier was pretreated with a NaCl solution of 300 mOsm/kg, equivalent to the osmolality of the PBS used. However, the different pretreatments led to no significant changes in the permeabilities of the two lipophilic model compounds, hydrocortisone and celecoxib.

Table 2 shows the permeation of the four compounds after pretreatment with NaCl solutions at 50 and 300 mOsm/kg across the PermeaPad^®^ barrier, the support layer, and the lipid layer. The permeation across the support layer of all four compounds was independent of the pretreatment. This is in contrast to the lipid layer, where the permeation of calcein and acyclovir was 2.0- and 1.7-fold higher when pretreating the barrier with the 50 mOsm/kg solutions.

The fact that only the permeability of the hydrophilic compounds across the lipid layer depends on the pretreatment shows that the pretreatment caused an alteration of the lipid layer inside the barrier.

Figure 2 illustrates our hypothesis schematically. When using a hypo-osmotic pretreatment solution compared to PBS, like the 50 and 150 mOsm/kg solutions, the lipid vesicles encapsulating the hypo-osmotic solution will form. Once the PBS replaces the pretreatment solution, water transfer out of the liposomes will occur and neutralize the osmotic difference between inside and outside the lipid vesicles, making the lipid vesicles shrink and leaving a larger space between the liposomes available for hydrophilic compounds to diffuse around the lipid vesicles.

While the hypothesis of liposomal shrinking upon hypo-osmotic pretreatment is well in line with our results, we expected the liposomes to swell and, thus, lower the permeability of calcein and acyclovir when pretreating the barrier with hyper-osmotic solutions. However, that was not the case, and, in fact, the permeability values of the hydrophilic compounds calcein and acyclovir increased significantly (*p* < 0.05) after the pretreatment of the barrier with the 900 mOsm/kg NaCl solution (Figure 1), which is in contrast to our expectations. We assume that the osmotic swelling might stress the lipid vesicles beyond the limit of flexibility and cause them to rupture and/or reorganize.

This study shows that investigators working with PermeaPad^®^ will have to be aware that differences in the osmolality of media used will alter the morphology of the barrier. In order to circumvent this, the addition of osmotically active compounds, such as NaCl or glucose, may be helpful to equalize osmolality. This is in contrast to studies with parallel artificial membrane permeability assays (PAMPA) that can be conducted with Pion’s acceptor sink buffer in the acceptor (~60 mOsm) and fed state-simulated intestinal fluid (~635 mOsm) in the donor [17] without affecting permeability.

The increased transport of hydrophilic compounds after shrinking the lipid vesicles within the barrier is assumed to be similar to paracellular transport. Although further studies with additional drug compounds are required, the results presented here indicate that this kind of barrier, consisting of lipid vesicles enclosed by two support sheets, may capture the paracellular transport of compounds, unlike several of the other existing non-cellular artificial permeation barriers, which are void of paracellular transport [18]. On the other hand, an artificial permeation model, which captures also the paracellular transport as well as transcellular transport, may bring about the advantage of higher intra- and inter-laboratory reproducibility and better cost and time efficiency than existing cellular models [8,19].

This study’s objective was to determine whether compounds passed the membrane in a paracellular-like manner and to test if liposome alteration by osmotic liposomal shrinking/swelling would change the permeation across the barrier. Future studies will find the most useful conditions for predicting the paracellular permeability of drug compounds. We still do not know if using the membrane with lipid vesicles of full or contracted size is most relevant, nor whether the PermeaPad^®^ barrier discriminates the paracellular-like transport of drug compounds in a biomimetic way. Improvement might be found in lipid composition changes or different types of support sheets.

### 3.2. The Time Necessary for Lipid Vesicles to Form

In all the previous listed experiments, the permeation barriers were pretreated in NaCl solutions for 30 min, a time frame that was randomly selected. We tested if we could cut down the time for pretreatment by performing additional experiments with calcein and the pretreatment of the barrier with 50 and 300 mOsm/kg NaCl solutions, but with pretreatment for just 2 min. The results are presented in Figure 3 and show no significant differences (*p* > 0.05) between the solutions pretreated for 2 and 30 min. The results show that the experiments are independent of the experimental time in the range of 2–30 min.

No previous studies have tested the time needed for the formation of lipid vesicles in the PermeaPad^®^. This experiment confirms that the lipid vesicles in the lipid layer of the barrier form within the first two minutes. This is an advantageous and required property of the barrier, as it typically does not come in contact with any liquids before starting the actual experiments.

## 4. Conclusions

In the current study, it was shown that the packing of liposomes within the PermeaPad^®^ barrier may be influenced by osmotic effects and that they are forming in less than two minutes. In consequence, the permeability of hydrophilic compounds, but not that of lipophilic compounds, was altered by the liposomal shrinking of the liposomes within the barrier. To the best of our knowledge, this is the first evidence of the modification of an artificial permeation barrier such that it does not just differentiate the permeability of drug compounds prone to pass the barrier by passive transcellular transport, but also via a paracellular-like transport mechanism.

## Figures and Tables

**Figure 1 pharmaceutics-14-00721-f001:**
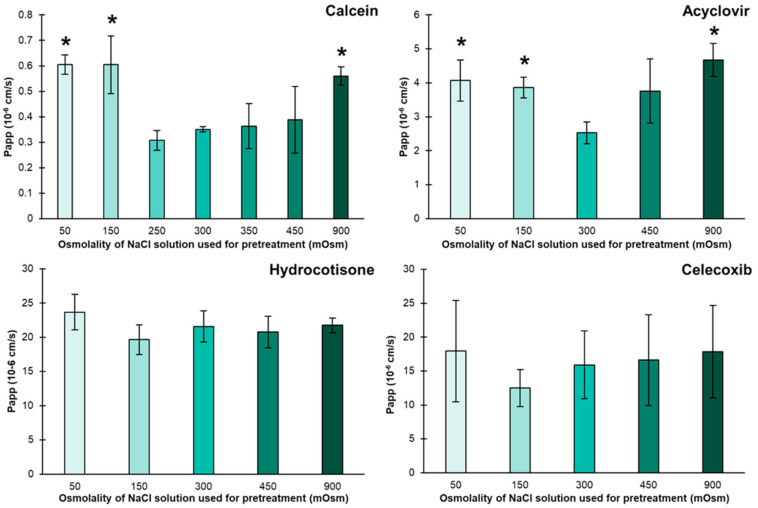
Permeation of calcein, acyclovir, hydrocortisone, and celecoxib across PermeaPad^®^ in 300 mOsm PBS after 30 min pretreatment of the barrier with NaCl solutions of different osmolalities. Mean ± SD (*n* = 3). * Indicates a significant difference (*p* < 0.05) as compared to the barriers pretreated with a 300 mOsm (isotonic) NaCl solution.

**Figure 2 pharmaceutics-14-00721-f002:**
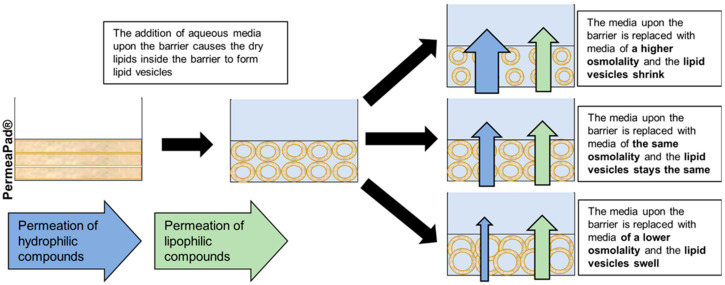
Schematic illustration of lipid vesicles shrinking due to the pretreatment of PermeaPad^®^ with a hypo-osmotic solution.

**Figure 3 pharmaceutics-14-00721-f003:**
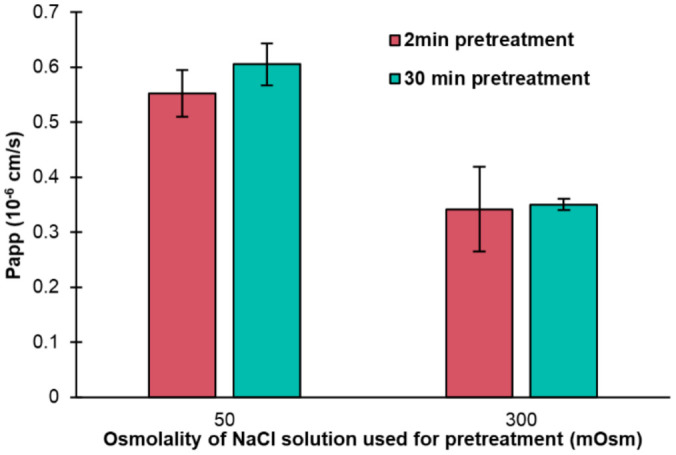
Permeation of calcein across PermeaPad^®^ in 300 mOsm PBS after 2 or 30 min pretreatment of the barrier with NaCl solutions at 50 (hypo-osmotic) or 300 (iso-osmotic) mOsm. Mean ± SD (*n* = 3).

**Table 1 pharmaceutics-14-00721-t001:** The data are obtained from chemicalize.com, using software by ChemAxon (Budapest, Hungary).

Compound	Molar Mass (g/mol)	Log D at pH 6.5	pKa (Strongest Acidic, Basic)	TPSA (Å2)	Solubility at pH 6.5 (mg/mL)
Calcein	622.55	−10.67	1.51 (8.15)	231.67	622.54
Acyclovir	225.21	−1.03	11.98 (3.02)	114.76	9.09
Hydrocortisone	362.47	1.28	12.59 (none)	94.83	0.41
Celecoxib	381.37	4.01	10.6 (0.41)	77.98	1.2·10^-3^

**Table 2 pharmaceutics-14-00721-t002:** Apparent permeability values of model compounds across PermeaPad^®^, the support layer, and the lipid layer.

	Calcein		Acyclovir		Hydrocortisone		Celecoxib	
Osmolality of NaCl solution used for pretreatment	50	300	50	300	50	300	50	300
Permeability across PermeaPad^®^ (10^−6^ cm/s)	0.61 ± 0.04	0.35 ± 0.01	4.07 ± 0.60	2.52 ± 0.33	23.68 ± 2.61	21.58 ± 2.27	17.95 ± 7.48	15.92 ± 4.99
Permeability across support layer (10^−6^ cm/s)	2.51 ± 0.30	2.84 ± 0.54	32.67 ± 1.87	33.19 ± 1.79	25.33 ± 2.77	23.69 ± 2.20	46.33 ± 2.23	46.16 ± 2.74
Permeability across lipid layer (10^−6^ cm/s)	0.80 ± 0.07	0.40 ± 0.02	4.71 ± 0.78	2.74 ± 0.38	N/A	N/A	34.16 ± 18.08	26.24 ± 11.24

## Data Availability

Data are fully contained within this article and in the Appendix A.

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
