# Peer review of "Modulation of Paracellular-like Drug Transport across an Artificial Biomimetic Barrier by Osmotic Stress-Induced Liposome Shrinking"

_pharmaceutics, 2022, doi:10.3390/pharmaceutics14040721_

Round 1

Reviewer 1 Report

The paper is pretty plain, and it is about a very important topic regarding pharmacological development. However, I would try to simplify the language because the readers might not be very super-expert in pharmacological testing. In particular, I would suggest to talk about osmolarity that is more common in the scientific field and I am still confused by this sentence:

While the hypothesis of liposomal shrinking upon hypo-osmotic pretreatment is well in line with our results, we expected the opposite when pretreating the barrier with hyper-osmotic solutions: swelling of lipid vesicles and thus lower permeability of calcein and acyclovir. However, that was not the case, and in fact, the permeability values of calcein and acyclovir increased significantly (p < 0.05) after pretreatment with the 900 mOsm/kg 
NaCl solution (Figure 1). We assume that the osmotic swelling might stress the lipid vesicles and cause them to rupture and/or reorganize.

To me, in hypoosmotic solution, the liposomes should swell and vice versa, but I might be wrong. When I tried to look for it, I found hypertonic and hypotonic that are more used and I suggest the authors to adopt this language in the paper. 

The paper shows the reactivity of the liposomes in contact with different saline solutions. They justify the behaviour of the membrane with some very rational thoughts about liposomes damage and shrinking. This is probably the case, but it is only a speculation. Taking only the liposomes and making different measurements of DLS and Z-pot with different saline solutions and with different drugs will increase the argument of the authors.

Author Response

Manuscript ID pharmaceutics-1638986

Modulation of paracellular-like drug transport across an artificial biomimetic barrier by osmotic stress-induced liposome shrinking

Jonas Borregaard Eriksen , Hesham Barakat , Barbara Luppi , Martin Brandl * , Annette Bauer-Brandl

Reviewer no 1 Comments and Suggestions for Authors

The paper is pretty plain, and it is about a very important topic regarding pharmacological development. However, I would try to simplify the language because the readers might not be very super-expert in pharmacological testing.

We have carefully revised the language of the whole manuscript and tried to eliminate or re-phrase sections, which may not be self-explaining for the reader.

In particular, I would suggest to talk about osmolarity that is more common in the scientific field and I am still confused by this sentence:

While the hypothesis of liposomal shrinking upon hypo-osmotic pretreatment is well in line with our results, we expected the opposite when pretreating the barrier with hyper-osmotic solutions: swelling of lipid vesicles and thus lower permeability of calcein and acyclovir. However, that was not the case, and in fact, the permeability values of calcein and acyclovir increased significantly (p < 0.05) after pretreatment with the 900 mOsm/kg 
NaCl solution (Figure 1). We assume that the osmotic swelling might stress the lipid vesicles and cause them to rupture and/or reorganize.

To me, in hypoosmotic solution, the liposomes should swell and vice versa, but I might be wrong. When I tried to look for it, I found hypertonic and hypotonic that are more used and I suggest the authors to adopt this language in the paper. 

Thank you for making us aware of this. There might be a misunderstanding; we are talking about hypo-osmotic pretreatment. In essence, what we mean with hypo-osmotic pretreatment is, that the dry lipids of the barrier were allowed to soak in a medium with lower osmotic pressure and when adding the test-solution containing the drug, the soaked barrier is then exposed to hypertonic stress, which let the vesicles shrink. We have tried to describe this in a more intuitive manner now in the abstract, specifying the terms. But in the manuscript itself we have refrained from replacing the terms hypo-osmotic and hyper-osmotic with hypotonic and hypertonic, respectively.

The paper shows the reactivity of the liposomes in contact with different saline solutions. They justify the behaviour of the membrane with some very rational thoughts about liposomes damage and shrinking. This is probably the case, but it is only a speculation. Taking only the liposomes and making different measurements of DLS and Z-pot with different saline solutions and with different drugs will increase the argument of the authors.

We are aware, that DLS-measurements likely would illustrate the change in liposome size, that is hypothesized. Unfortunately, the barrier comes as a sandwich, where the (dry) lipids are embedded within two support sheets (dialysis membranes). The support sheets cannot easily be removed to release the liposomes formed in the space inbetween, at least not without exerting mechanical stress, which in turn would influence liposome size. We thus cannot imagine a (straightforward) approach to measure liposome size within the barrier. We have added a literature reference where the shrinking and swelling of liposomes due to osmotic pressure differences has been experimentally studied (Judy Senior).

Reviewer 2 Report

Dear Authors,

The manuscript entitled "Modulation of paracellular-like drug transport across an artificial biomimetic barrier by osmotic stress-induced liposome shrinking" investigates the transport of different drug models through a biomimetic membrane pre-treated by different osmolalities. The findings suggested that osmolality affected the lipid vesicles packed in the membrane which in turn, had an impact on the permeation rate of the hydrophilic drugs. 

The manuscript provides new insights about passive diffusion; it is well-written and well-organized, as well as, clear and concise in general. I only have a few comments:

  1. The introduction is contextualized in the oral route and intestinal absorption, then the sentence about human skin as predictive paracellular transport (lines 50-51) looks to be out of place. In my opinion, I would delete it.
  2. Section 2.2. Media preparation: it is unclear for me the role of the liposomes described in this section. The reader does not figure out that they were packed in the membranes until the conclusions section. One guesses it, as the manuscript progresses and it is finally stated in the conclusions. I think it should be clarified:

I assume: Different osmolality liposomes were prepared and packed in the membranes. And additionally, the biomimetic membranes were pre-treated with NaCl solutions at different osmolalities as well (lines 144-147) (?). It is unclear to me how many different membranes were used and if all of them were exposed to all of the NaCl solutions. Please clarify. 

3. Figure 1:  There is a typo error in the legend (line 214). 

4. Figure 1: shows 7 levels of osmolalities for Calcein and only 5 levels for the rest Acyclovir, Hydrocortisone, and Celecoxib. Please include the missing osmolalities, or exclude 250 and 350 mOsm in the Calcein plot; so, the plots are presented in a unified format. *** If the latter option is selected, review then line 119. 

Kind regards,

Author Response

Manuscript ID pharmaceutics-1638986

Modulation of paracellular-like drug transport across an artificial biomimetic barrier by osmotic stress-induced liposome shrinking

Jonas Borregaard Eriksen , Hesham Barakat , Barbara Luppi , Martin Brandl * , Annette Bauer-Brandl

Reviewer no 2

The manuscript entitled "Modulation of paracellular-like drug transport across an artificial biomimetic barrier by osmotic stress-induced liposome shrinking" investigates the transport of different drug models through a biomimetic membrane pre-treated by different osmolalities. The findings suggested that osmolality affected the lipid vesicles packed in the membrane which in turn, had an impact on the permeation rate of the hydrophilic drugs. 

The manuscript provides new insights about passive diffusion; it is well-written and well-organized, as well as, clear and concise in general. I only have a few comments:

  1. The introduction is contextualized in the oral route and intestinal absorption, then the sentence about human skin as predictive paracellular transport (lines 50-51) looks to be out of place. In my opinion, I would delete it.

Maybe this is a mistake? We could not identify the sentence about human skin. But in order to avoid misunderstandings, we have modified the section to specify the whole discussion is about intestinal permeation.

  1. Section 2.2. Media preparation: it is unclear for me the role of the liposomes described in this section. The reader does not figure out that they were packed in the membranes until the conclusions section. One guesses it, as the manuscript progresses and it is finally stated in the conclusions. I think it should be clarified:

We added a sentence for clarification in the Media preparation section.

I assume: Different osmolality liposomes were prepared and packed in the membranes. And additionally, the biomimetic membranes were pre-treated with NaCl solutions at different osmolalities as well (lines 144-147) (?). It is unclear to me how many different membranes were used and if all of them were exposed to all of the NaCl solutions. Please clarify. 

Section was thoroughly revised / re-written to clarify.

  1. Figure 1:  There is a typo error in the legend (line 214). 

Corrected.

  1. Figure 1: shows 7 levels of osmolalities for Calcein and only 5 levels for the rest Acyclovir, Hydrocortisone, and Celecoxib. Please include the missing osmolalities, or exclude 250 and 350 mOsm in the Calcein plot; so, the plots are presented in a unified format. *** If the latter option is selected, review then line 119. 

After thorough consideration, we have decided to keep the figure as it is. In the first experiment with Calcein, we had learned, that the osmolality-effect is well illustrated by 5 of the 7 levels and therefore omitted two of the levels in subsequent experiments. Still, we do not see a need to remove these two extra levels from the Calcein panel.

Round 2

Reviewer 1 Report

NA